# IgA autoantibodies promote inflammation, Th17 polarization and fibrotic responses in hidradenitis suppurativa

Hidradenitis suppurativa (HS) is a chronic inflammatory skin disease with complex immune dysregulation. While previous studies have reported tertiary lymphoid structures (TLS) and IgA-producing B cells in HS skin lesions, the pathogenic role of IgA autoantibodies remains insufficiently characterized. Here, we demonstrate that lesional skin from patients with HS exhibits elevated expression of IGHA1, IGHA2, and J chain, as confirmed by qPCR, immunofluorescence, and Western blot analyses. Single-cell RNA sequencing identified local plasma cells and B cells as the primary source of IgA. Autoantigen profiling revealed a diverse repertoire of IgA autoantibodies targeting nuclear, cytoplasmic, and membrane antigens, with levels correlating with disease severity and other clinical manifestations. We identified IgA autoantibodies binding to CD68+ macrophages, which induced secretion of TNF, IL-6, and IL-1β and upregulated inflammasome and profibrotic pathways. Anti-neutrophil extracellular trap (NET) IgA was elevated in HS and promoted NET formation, establishing a pathogenic feedback loop. NET−IgA immune complexes induced macrophages to secrete CCL18, driving collagen production by fibroblasts and promoting a type I interferon (IFN) gene signature. IgA immune complexes presented by myeloid dendritic cells activated CD4+ T cells, triggering IFN-γ production and further amplifying local inflammation. Notably, supernatants from macrophages exposed to IgA were able to polarize naïve CD4+ T cells toward a Th17 phenotype, linking innate immune activation to the expansion of pathogenic adaptive immune responses. Direct exposure of fibroblasts to NET−IgA complexes triggered expression of adhesion molecules, chemokines, and regulators of adaptive immunity. Together, these findings uncover a role for IgA autoantibodies in HS, implicating them as central drivers of chronic inflammation, fibrosis, and immune crosstalk across neutrophils, macrophages, fibroblasts, and T cells.

Hidradenitis suppurativa (HS) is a chronic, relapsing inflammatory skin disorder characterized by painful nodules, abscesses, and draining tunnels, most commonly affecting intertriginous regions such as the axillae, groin, and buttocks[1–3]. Despite increasing recognition of its multifactorial pathogenesis, including genetic susceptibility, follicular occlusion, and immune dysregulation, the underlying immunologic mechanisms driving HS remain incompletely understood[4–11]. While early studies emphasized the role of neutrophil-mediated

✉e-mail: carmelo.carmona-rivera@nih.gov

inflammation and complement activation, recent transcriptomic and histopathologic analyses have also identified robust infiltration of B cells and plasma cells within HS lesions, suggesting a significant contribution of humoral immunity to disease pathophysiology[5,7,9,12–15].

One particularly intriguing finding has been the presence of tertiary lymphoid structures (TLS)[9,15,16]. These organized lymphoid aggregates, composed of T cells, B cells, and follicular dendritic cells (FDCs), are known to serve as local sites of antigen presentation, B cell activation, and antibody production in chronic inflammatory diseases[9,15,17,18]. In HS, their presence raises the possibility that in situ antibody generation may play a more central role than previously appreciated. Indeed, autoantibodies, including IgG reactive to citrullinated proteins, have been detected in patient sera and lesional tissue, and B cell/plasma cell signatures are increasingly recognized in HS transcriptomic profiles[5,7,13,14].

However, while IgG autoantibodies have received some attention, the potential role of immunoglobulin A (IgA), a mucosal isotype with both immunoregulatory and pro-inflammatory properties, remains poorly characterized in HS. Emerging evidence indicates local class switching to IgA in HS lesions[7,9,15]. However, it remains unclear whether IgA autoantibodies are produced in HS skin lesions, what antigens they target, and what functional effects they exert within the inflamed skin microenvironment. Given the ability of IgA immune complexes to activate myeloid[19] and stromal cells via Fcα receptors[20], and the established presence of TLSs as niches supporting local B cell maturation, the possibility that IgA contributes to sustained inflammation and tissue remodeling in HS warrants focused investigation.

This study aims to address this critical gap by investigating the cellular sources, antigenic targets, and downstream immunologic effects of IgA in HS. By characterizing the expression of IgA-related genes and proteins, profiling the specificity of IgA autoantibodies, and examining their impact on key immune and resident skin cells, we seek to clarify how IgA contributes to the chronic inflammation, fibrosis, and immune amplification that define advanced HS.

## Results

### IgA antibodies are present in HS skin lesions

Histological analysis of skin lesions from patients with HS revealed organized lymphoid aggregates consistent with TLSs[9,15,17]. To investigate local IgA production, we performed quantitative PCR on skin biopsies from healthy controls and HS patients. Expression of *JCHAIN* and *IGHA1* was significantly elevated in HS lesions compared to control skin (Fig. 1A, B, Suppl Fig. S1A, B; $p = 0.0368$). Similarly, ELISA analysis demonstrated significantly increased IgA protein levels in HS lesional tissue ($p = 0.0013$) but not in serum (Fig. 1C, D, Suppl Fig. s1C, D). To determine whether IgA elevation extended beyond lesional areas, we analyzed paired lesional and non-lesional skin samples from publicly available RNA sequencing data obtained from the same patients[21]. Lesional tissue exhibited significant upregulation of *JCHAIN* ($p = 0.0004$) and *IGHA1* ($p < 0.0001$), whereas non-lesional sites showed minimal or no expression (Fig. 1E, F). Immunofluorescence staining confirmed IgA protein localization throughout lesional tissue, with prominent (Fig. 1G). Western blot analysis of lesional extracts further validated elevated levels of J chain, IgA1, and IgA2 proteins in HS skin compared to healthy controls (Fig. 1H). IgA2 (heavy chain [Hc] and light chain [Lc]) appears to be more prevalent in HS skin lesions than the IgA1 subtype (Fig. 1H). To identify the cellular source of IgA in skin, we analyzed publicly available single-cell RNA sequencing (scRNA-seq) data from lesional HS skin[9]. This analysis revealed that B cells and plasma cells expressing *IGHA1*, *IGHA2*, and *JCHAIN*, consistent with local class-switch recombination and in situ IgA production (Fig. 1I). To further substantiate that B cells contribute to the IgA detected in HS skin lesions, we performed confocal microscopy analysis. Immunofluorescence revealed IgA-producing cells clustered

within CD20$^+$/CD21$^+$ B cell-rich areas, consistent with TLS-like aggregates (Fig. 1J). Notably, IgA-producing cells were also detected outside these aggregates and distributed throughout the tissue. Collectively, these findings demonstrate the presence of IgA-producing plasma cells within HS lesions, frequently in proximity to B cell clusters, supporting a localized, tissue-specific humoral immune response in HS.

### IgA autoantibodies are present in HS and correlate with clinical manifestations

Given the evidence for local IgA production in HS lesions, we next investigated whether these antibodies were directed against self-antigens. Using comprehensive autoantigen profiling across two independent HS patient cohorts, we identified a broad and diverse repertoire of IgA autoantibodies targeting nuclear (e.g., dsDNA, nucleolin, genomic DNA, PCNA), cytoplasmic (e.g., vimentin, MDA5, complement components), and extracellular antigens (e.g., collagen, fibronectin, IFN-γ) (Fig. 2A, B). Many of these autoantibodies were more prevalent in HS patients with chronic or severe disease. The demographic and clinical characteristics of the cohorts are summarized in Table 1.

Notably, IgA autoantibodies against melanoma differentiation-associated protein 5 (MDA5; $p = 0.002$), thyroglobulin ($p = 0.0022$), complement proteins C3 ($p = 0.0023$) and C4 ($p = 0.0079$), NLR family pyrin domain containing 1 (NLRP1; $p = 0.0182$), and angiotensin-converting enzyme (ACE; $p = 0.002$) were significantly elevated in patients with advanced disease (higher Hurley stage), suggesting an association between these IgA autoantibodies and disease progression and tissue injury. Conversely, IgA autoantibodies targeting myeloperoxidase (MPO; $p = 0.0042$), IL-17A ($p = 0.0122$), and nucleolin ($p = 0.0198$) were more frequently observed in patients with milder disease or in healthy controls, indicating potential context-dependent regulatory functions (Fig. 2C). Correlation analyses further revealed associations between specific IgA autoantibodies and clinical manifestations. For example, IgA antibodies to aggrecan ($r = 0.4505$, $p = 0.0231$), calprotectin ($r = 0.4560$, $p = 0.0226$), and collagen II ($r = 0.4719$, $p = 0.0178$) were significantly associated with acne (Fig. 2D). Diabetes showed significant correlations with IgA against nuclear antigens, including Ro/SSA (52 kDa) ($r = 0.7299$, $p = 0.0001$), Sm/RNP ($r = 0.4994$, $p = 0.0125$), and small nuclear ribonucleoprotein SmD3 ($r = 0.6332$, $p = 0.0013$).

Disease severity, as measured by Hurley stage, was positively associated with IgA autoantibodies to MDA5 ($r = 0.4997$, $p = 0.0146$), genomic DNA ($r = 0.4800$, $p = 0.0187$), and dsDNA ($r = 0.4372$, $p = 0.0306$). The number of draining sinuses correlated with anti-calprotectin ($r = 0.4416$, $p = 0.0256$) and proteinase-3 (PR3) IgA ($r = 0.4749$, $p = 0.0171$), while anti-alpha fodrin IgA was associated with nodule count ($r = 0.4011$, $p = 0.0398$), local IHS4 score ($r = 0.4713$, $p = 0.0179$), and number of draining sinuses ($r = 0.4994$, $p = 0.0125$).

Smoking, an established HS risk factor, was significantly associated with IgA responses to laminin ($r = 0.6104$, $p = 0.0021$), complement C1q ($r = 0.6103$, $p = 0.0021$), and histone H2B ($r = 0.4590$, $p = 0.0209$). The presence of comedones was linked to IgA targeting Sm ($r = 0.6380$, $p = 0.0012$), glomerular basement membrane (GBM; $r = 0.4135$, $p = 0.0349$), IFN-γ ($r = 0.5377$, $p = 0.0072$), lysozyme C ($r = 0.4315$, $p = 0.0287$), and nucleosomes ($r = 0.4031$, $p = 0.0390$).

Of note, several IgA autoantibodies were inversely associated with the use of anti-TNF therapy, including those against BAFF ($r = -0.4125$, $p = 0.0353$), CD40 ($r = -0.3936$, $p = 0.0429$), collagen IV ($r = -0.4379$, $p = 0.0267$), La/SSB ($r = -0.3884$, $p = 0.0452$), MDA5 ($r = -0.4560$, $p = 0.0216$), myosin ($r = -0.4088$, $p = 0.0367$), and PR3 ($r = -0.5194$, $p = 0.0094$) (Fig. 2D).

Because IgA derived from mucosal and tissue compartments can exhibit a degree of polyreactivity[22], we next assessed whether the IgA detected in HS lesions recognized bacterial antigens and might

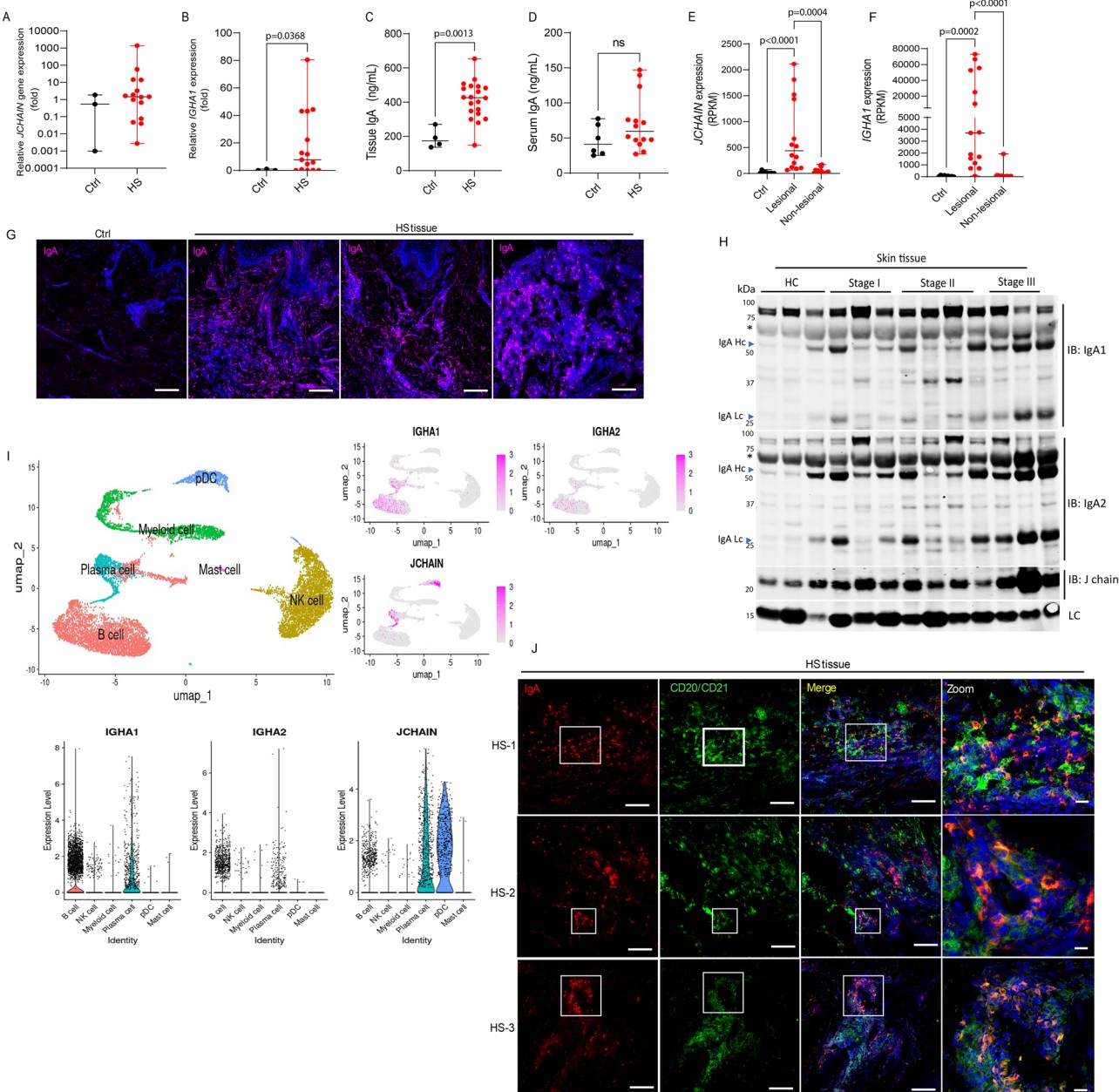

**Fig. 1 | IgA expression is elevated in Hidradenitis Suppurativa (HS) skin lesions.**
**A** qPCR analysis of *JCHAIN* and **B** *IGHA1* expression in lesional HS skin (*n* = 15) compared to control tissue (Ctrl, *n* = 3); **C, D** ELISA quantification of total IgA levels in HS versus control skin (Ctrl *n* = 4, HS *n* = 20) and serum samples (Ctrl *n* = 6, HS *n* = 14). **E, F** Expression of *JCHAIN* and *IGHA1* in lesional (*n* = 15), non-lesional (*n* = 15), and control (*n* = 9) skin samples from a publicly available transcriptomic dataset (GSE155176). **G** Representative confocal images of skin lesions from HS patients and controls stained for IgA (magenta) and DNA (Hoechst, blue); Scale bar 100 μm. Images are representative of 6 HS patients and 3 controls. **H** Western blot analysis of homogenized skin lysates from HS patients across different Hurley stages and healthy controls (HC) showing protein levels of IgA1, IgA2, and J chain; Lc = loading control, * indicates non-specific bands. Images are representative of three independent experiments. **I** UMAP clustering of skin single-cell RNA sequencing data from HS lesions (GSE249793) showing *IGHA1, IGHA2*, and *JCHAIN* expression in B cells, plasma cells, and DCs. **J** Confocal images of skin lesions from three HS patients stained for B cell (CD20/CD21, green), IgA (red) and DNA (Hoechst, blue); Scale bar 100 μm, zoom 10 μm. Data are presented as median ± range of three independent experiments in duplicate. For the statistical analysis, a 2-sided unpaired Mann-Whitney *U*-test was used for (**A–D**), and a Kruskal-Wallis test was used in (**E, F**). Source data are provided as a Source data file.

therefore confound our interpretation. ELISA analyses of IgA reactivity against *E. coli*, *Lactobacillus*, and *P. gingivalis* revealed no significant differences between HS tissue samples and controls (Fig. 2E, Suppl Fig. S2A–C). In addition, correlation analyses showed no positive association between anti-bacterial IgA levels and the clinical parameters examined (Fig. 2F). Notably, anti–*E. Coli* IgA levels in HS samples positively correlated with draining sinus count, mirroring the only autoantibody that correlated with this parameter, anti-CRP IgA. This observation suggests possible cross-reactivity or a shared

inflammatory context rather than a direct microbe-driven response. Although total IgG levels were significantly elevated in HS samples (Supplementary Fig. S1E, F), IgG levels correlated with male sex but not with other clinical parameters (Fig. 2F), further supporting a distinct functional role for IgA in HS pathogenesis. These findings suggest that IgA autoantibodies in HS display a dynamic and heterogeneous repertoire, with both potentially pathogenic and regulatory roles. Their distribution appears to be shaped by disease severity, clinical phenotype, and therapeutic exposure.

**Table 1 | Demographics of the patients with HS tested in this study**

| Demographic | Cohort 1 ($n = 25$) | Cohort 2 ($n = 20$) | Ctrl ($n = 8$) |
|---|---|---|---|
| Age, median (IQR) | 39 (17) | 34 (12) | N/A |
| Female sex, n (%) | 18(72) | 14 (70) | 8 (100) |
| **Race** | | | |
| Black n (%) | 17 (68) | 10 (50) | 4 (50) |
| White n (%) | 5 (20) | 8 (40) | 4 (50) |
| Other n (%) | 3 (12) | 2 (10) | |
| **Smoking history** | | | |
| Current use, n (%) | 4 (16) | 1 (5) | N/A |
| Previous smoker, n (%) | 6(24) | 3 (15) | N/A |
| Never smoked, n (%) | 15 (60) | 16(80) | N/A |
| **BMI**, median (IQR) | 37 (11) | 35 (13) | N/A |

Abbreviations: *BMI* body mass index; *IQR* interquartile range.

## HS-IgA facilitates keratinocyte autoantigen presentation through FcαRI and co-stimulatory molecules

To further characterize the IgA antibody repertoire in HS, we performed Western blot analysis using lysates from control neutrophils, skin fibroblasts, and keratinocytes (HaCaT cells). Notably, IgA2 autoantibodies purified from HS skin samples robustly recognized antigens from all three cell types, whereas minimal reactivity was observed in control samples (Fig. 3A, upper panel), suggesting the presence of IgA2 antibodies recognizing autoantigens in neutrophils, skin fibroblasts and keratinocytes in HS skin lesions. In contrast, IgA1 antibodies from healthy control and HS skin showed no difference in their ability to recognize autoantigens in these 3 cell types (Fig. 3A, lower panel), indicating that IgA2 is the predominant subtype targeting skin and immune cells in HS. To investigate the functional consequences of IgA autoreactivity, IgA was purified from HS lesional tissue using protein M. We hypothesized that IgA bound to keratinocyte antigens could form immune complexes capable of being presented to CD4$^+$ T cells via myeloid DCs (mDCs), contributing to local immune activation. To test this, we isolated autologous CD4$^+$ T cells and mDCs from healthy donors and co-cultured them in the presence or absence of keratinocyte protein–IgA immune complexes for 4–5 days. IFN-γ and TNF-α secretion were used as a readout for T cell activation. Co-cultures exposed to keratinocyte–IgA complexes exhibited significantly elevated levels of both cytokines (IFN-γ, $p = 0.0265$; TNF-α, $p = 0.0001$), indicating effective T cell activation (Fig. 3B, C). To determine the mechanism of autoantigen uptake and presentation, we used neutralizing antibodies against the IgA receptor FcαRI (CD89) and co-stimulatory molecules CD80/CD86. Blocking these pathways significantly reduced IFN-γ production ($p = 0.0347$, $p = 0.0011$) (Fig. 3D), suggesting that keratinocyte–IgA immune complexes are internalized via CD89 and require co-stimulation for T cell activation. These findings support a model in which IgA autoantibodies facilitate antigen uptake and presentation by mDCs, leading to activation of autoreactive CD4$^+$ T cells and promoting IFN-γ/TNF-α–driven skin inflammation. This mechanism highlights a potential bridge between innate and adaptive immunity in HS.

## IgA autoantibodies target macrophages and promote a pro-inflammatory and pro-fibrotic response in HS

Building on the discovery of a broad IgA autoantibody repertoire in HS, we identified a specific subset of IgA autoantibodies recognizing macrophage proteins that were significantly elevated in HS skin ($p < 0.0001$) compared to control skin (Fig. 4A) and positively associated with disease severity (r = 0.3845, $p = 0.0238$; Fig. 4B). Immunofluorescence staining of HS lesions revealed co-localization of IgA with CD68$^+$ cells, supporting the presence of IgA-bound resident or

infiltrating macrophages in lesional tissue (Fig. 4C). To determine whether these antibodies specifically recognize macrophage antigens, we performed ELISA using recombinant CD68. HS samples exhibited significantly higher levels of anti-CD68 IgA ($p = 0.0026$) compared to controls (Fig. 4D), with a strong correlation between antibody levels and disease severity (r = 0.4671, $p = 0.0189$; Fig. 4E). To assess the functional consequences of IgA binding, we exposed primary human M2-like macrophages to IgA purified from HS lesions, control IgA, or vehicle for 24 h. RNA sequencing of macrophages treated with HS IgA revealed substantial upregulation of genes involved in inflammation, leukocyte chemotaxis, fibroblast activation, and the NLRP3 inflammasome pathway (Fig. 4F, G). ELISA of culture supernatants confirmed significantly increased secretion of TNF-α ($p = 0.0308$), IL-6 ($p = 0.0002$), and IL-1β ($p = 0.0169$) by macrophages treated with HS IgA compared to control IgA or untreated macrophages (Fig. 4H–J). Given our previous observation of IgG deposition in HS skin lesions[13], we next assessed whether IgG purified from the same HS tissues elicited proinflammatory responses comparable to those induced by IgA. IgG isolated from HS samples induced significant release of TNF-α ($p = 0.0047$), and IL-6 ($p = 0.0016$), but did not stimulate significant IL-1β production (Fig. 4H–J), highlighting a functional divergence between IgG- and IgA-mediated macrophage activation.

Given the cytokine profile, we hypothesized that HS IgA–activated macrophages could promote Th17 polarization. To test this hypothesis, naïve CD4$^+$ T cells from healthy donors were purified and cultured with 10% supernatant from macrophages treated with either HS-IgA or control IgA. After 4 days, qPCR revealed significant upregulation of *RORC* ($p = 0.0055$), encoding the Th17 lineage-defining transcription factor RORγt, in T cells exposed to HS IgA-conditioned media (Fig. 4K). Additionally, IL-17A ($p = 0.0005$) levels were significantly elevated in these cultures (Fig. 4L), indicating that macrophage-derived factors driven by HS IgA promote Th17 differentiation. Consistent with these findings, supernatants from macrophages treated with HS-derived IgG failed to promote polarization of naïve T cells toward a Th17 phenotype, as evidenced by significantly lower IL-17A production (p = 0.0025) compared with naïve T cells exposed to supernatants from macrophages stimulated with HS-derived IgA (Fig. 4L). Together, these results indicate that IgA-but not IgG- autoantibodies in HS preferentially activate macrophages to produce Th17-polarizing cytokines, thereby linking humoral autoimmunity to pathogenic adaptive immune responses in the skin.

## IgA autoantibodies recognize NETs and amplify inflammation and fibrosis via macrophage and fibroblast crosstalk

Further expanding the spectrum of IgA autoreactivity in HS, we identified significantly elevated levels of IgA antibodies directed against neutrophil extracellular traps (NETs, $p = 0.0079$) in lesional HS skin compared to controls (Fig. 5A). This was relevant as NETs have been implicated in HS pathogenesis[5,14]. These anti-NET antibodies were significantly elevated in Hurley stage III when compared to controls ($p = 0.0057$) (Fig. 5B). Immunofluorescence staining of HS tissue revealed co-localization of IgA and neutrophil elastase, a hallmark component of NETs, supporting the presence of IgA autoantibodies bound to NET structures in vivo (Fig. 5C). To assess whether IgA can modulate NET formation, control neutrophils were incubated with purified IgA from HS or control skin. Indeed, HS IgA induced significant NET release ($p = 0.0047$) when compared to control IgA (Fig. 5D). These findings suggest a vicious cycle, in which IgA antibodies recognize and stimulate NET formation, potentially perpetuating local inflammation.

To explore the impact of NET–IgA immune complexes on other cell types involved in HS pathogenesis, we incubated M2 polarized macrophages with NETs alone or with NET–IgA complexes. Supernatant analysis revealed that NET–IgA complexes induced significant release of CCL18 ($p = 0.0022$) (Fig. 5E), a chemokine known to stimulate collagen synthesis in dermal fibroblasts and proposed to play a pathogenic role in HS. Of note, IgA alone did not induce significant

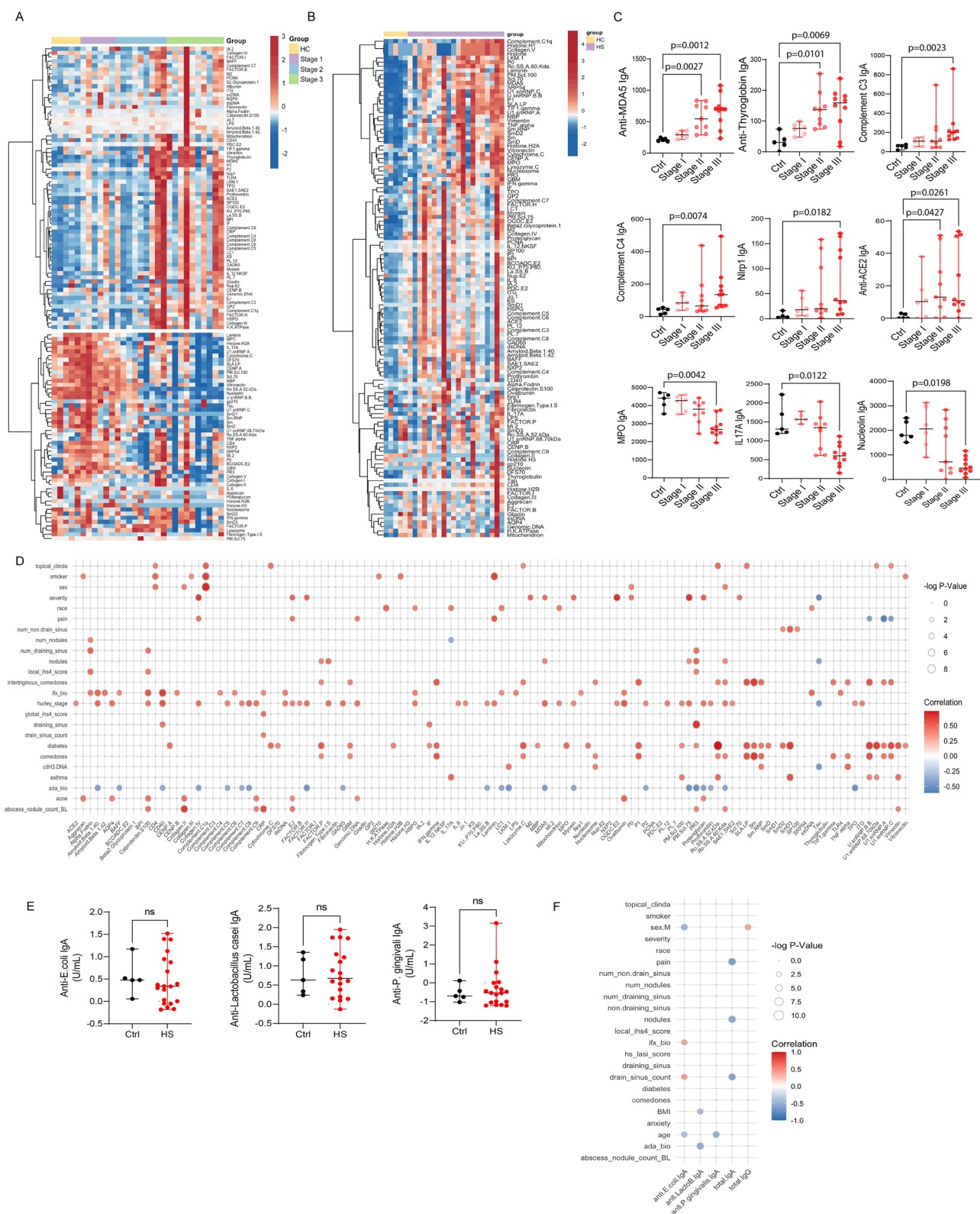

CCL18 release (Suppl Fig. S3). To determine the downstream impact of this macrophage-derived signal, primary skin fibroblasts were incubated with supernatants from M2 macrophages treated with NETs or NET–IgA complexes and RNA sequencing was performed. Notably, skin fibroblasts treated with NET–IgA supernatant exhibited an upregulated type I interferon (IFN) gene signature, as well as significant upregulation of pro-inflammatory genes including *IL6, CXCL3, CCL2*, and *IL1B* (Fig. 5F, G). ELISA analysis showed that fibroblasts exposed to

NET–IgA-conditioned M2 supernatants produced significantly elevated levels of collagen when compared to those exposed to NETs alone (*p* = 0.0317), suggesting that NET–IgA immune complexes may drive fibroblast-mediated fibrosis via macrophage signaling. These results suggest that NET–IgA immune complexes not only promote fibrosis but also amplify innate immune activation in fibroblasts.

To assess whether NET–IgA complexes can act directly on fibroblasts, these cells were also incubated with purified NETs or NET–IgA

**Fig. 2 | IgA autoantibodies are present in HS skin lesions and correlate with disease severity. A**, **B** Heatmaps show unsupervised clustering of IgA auto-antibodies detected in skin tissue from two independent HS cohorts: **A** cohort 1 (Ctrl, $n = 5$; HS, $n = 25$) and **B** cohort 2 (Ctrl, $n = 5$; HS, $n = 20$). **C** Skin lysates from controls ($n = 5$) and HS patients (stage I: $n = 6$; stage II: $n = 9$; stage III: $n = 11$) were analyzed for IgA autoantibodies targeting MDA5, thyroglobulin, complement components C3 and C4, NLRP1, ACE, MPO, IL-17A, and nucleolin. **D** Correlation analysis of IgA autoantibody levels in HS skin lesions with clinical outcomes ($n = 20$), with the color gradient indicating correlation coefficients (R values). ada:

adalimumab; ifx: infliximab; pain: numeric pain rating score; severity: self-reported severity. **E** ELISA-based quantification of IgA antibodies against *E. coli*, *Lactobacillus casei*, and *P. gingivalis* in HS skin lesions ($n = 20$) and controls ($n = 5$). **F** Correlation analysis between total IgA, total IgG, and IgA antibodies against *E. coli*, *Lactobacillus casei*, and *P. gingivalis* in HS skin lesions and clinical outcomes ($n = 20$). The data are shown as Z-score heatmaps in (**A**, **B**), median ± range in (**C**, **E**), and correlation coefficients in (**D**, **F**). For the statistical analysis, the Kruskal-Wallis test was used in (**C**, **E**). Source data are provided as a Source data file.

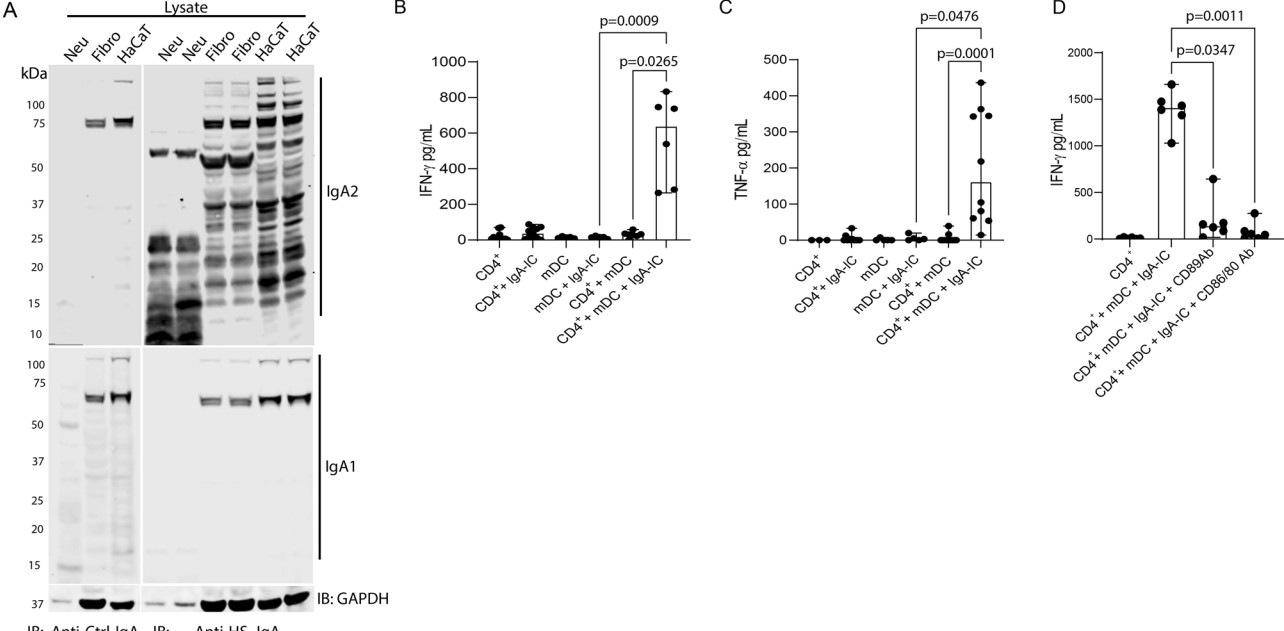

**Fig. 3 | IgA facilitates keratinocyte antigen uptake by mDCs via FcαR and promotes CD4⁺ T cell activation. A** Protein lysates from control neutrophils (Neu), skin fibroblasts (Fibro), and keratinocytes (HaCaT) were separated by SDS-PAGE and transferred to nitrocellulose membranes. Membranes were probed with 10 μg of skin lysate from control ($n = 5$) or HS patients ($n = 5$) as a source of skin-derived IgA. Autoantibodies were detected using anti-IgA1 or anti-IgA2 secondary antibodies. GAPDH was used as a loading control. Images are representative of three independent experiments. **B**, **C** Detection of IFNγ (**B**) and TNFα (**C**) in co-cultures of control mDCs and autologous CD4⁺ T cells in the presence or absence of

keratinocyte–IgA immune complexes (IgA-IC) after 5 days. The data are presented as median ± range of five independent experiments in duplicate. The Kruskal-Wallis test was used. **D** Detection of IFNγ following co-culture of mDCs with autologous CD4⁺ T cells and keratinocyte–IgA immune complexes, in the presence or absence of neutralizing antibodies targeting FcαR (CD89) or the co-stimulatory molecules CD80/CD86. The data are presented as median ± range of six independent experiments in duplicates. For the statistical analysis, the Kruskal-Wallis test was used. Source data are provided as a Source data file.

complexes. RNA-seq analysis revealed that NET–IgA-treated fibroblasts upregulated genes involved in leukocyte recruitment and adhesion, such as *CD9, ITGA2, VCAM1, ICAM1*, and those encoding for several CXCL chemokines, consistent with a pro-inflammatory and pro-chemotactic phenotype (Fig. 5I). Additionally, genes associated with adaptive immune modulation, including *TNFSF13B* (BAFF), *TNFAIP3*, and *NFKBIZ*, were significantly upregulated (Fig. 5J), suggesting that NET–IgA complexes may also prime fibroblasts to act as immune-competent stromal cells that influence local adaptive responses. These findings support a model in which NET–IgA immune complexes serve as pleiotropic and potent inflammatory amplifiers, capable of activating macrophages and fibroblasts in a way that bridges innate immunity, tissue remodeling, and adaptive immune regulation, key processes that underlie the chronic and fibrotic nature of HS lesions.

## Discussion

While innate immune components such as neutrophils and complement have long been implicated in HS pathogenesis, growing evidence highlights a significant role for adaptive immunity, particularly B cells

and plasma cells. Previous studies reported TLSs and B cell–related gene signatures in HS lesions[7,9]; yet the functional relevance of these cells remained unclear. Our findings reveal a central role for IgA-producing B cells and IgA autoantibodies in HS. We show that lesional skin exhibits robust local IgA production, with elevated expression of *IGHA1*, *IGHA2*, and *JCHAIN* compared to non-lesional and healthy skin. Single-cell RNA sequencing confirmed that B cells and plasma cells are the primary IgA source, indicating HS lesions are functionally active and support class switching and antibody production.

B-cell hyperactivity, elevated BAFF, and in situ immunoglobulin production in HS mirror mechanisms observed in autoimmune diseases like SLE and RA[5,7,9,13,14], where B cells contribute to disease pathogenesis via autoantibody generation and immune complex formation. TLSs in HS lesions parallel those seen in RA synovium and lupus nephritis, suggesting chronic, localized immune activation[23–25]. In HS, IgA-producing B cells generate a diverse repertoire of autoantibodies targeting nuclear, cytoplasmic, and surface antigens such as CD68. Several IgA reactivities correlate with disease severity (e.g., Hurley stage, nodule count, sinus formation), and the repertoire appears functionally heterogeneous, with some IgA species

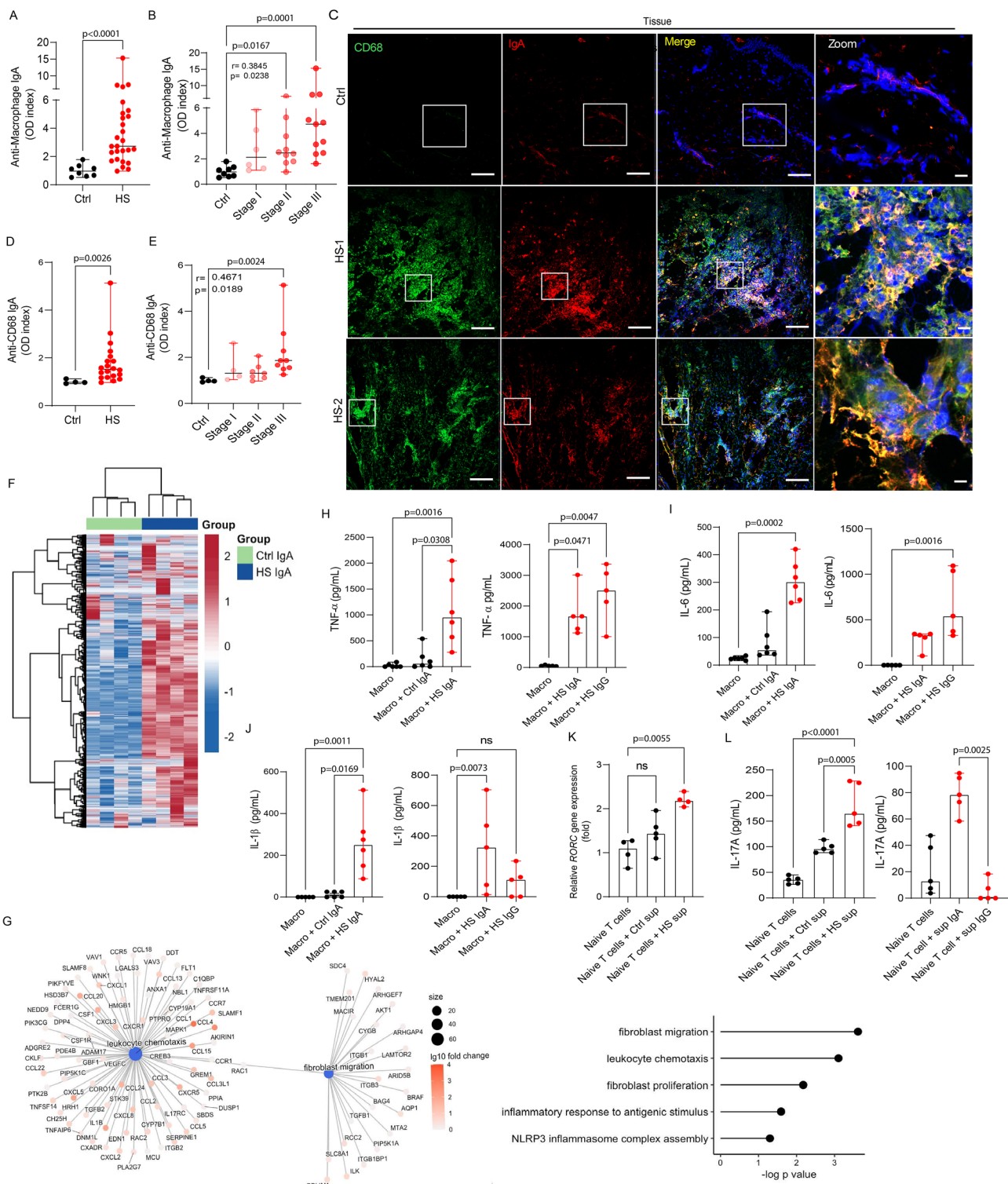

**Fig. 4 | IgA antibodies activate macrophages and promote Th17 polarization.**
**A** Anti-macrophage (control (Ctrl, $n = 8$) and HS patient samples ($n = 27$)) and
**D** anti-CD68 IgA antibody levels measured in homogenized lysates from control
(Ctrl, $n = 4$) and HS patient samples ($n = 20$). The data are presented as median ±
range of three independent experiments in duplicates. For the statistical analysis, a
2-sided unpaired Mann-Whitney U-test was used. **B**, **E** Correlation of anti-
macrophage **B** and anti-CD68 **E** IgA antibody levels with HS severity based on
Hurley stage. For the statistical analysis, the Kruskal-Wallis test was used.
**C** Representative confocal images of skin lesions from HS patients and controls
(ctrl) stained for CD68 (green), IgA (red), and DNA (Hoechst, blue). Images are
representative of 6 HS patients and 3 controls. Scale bars, 100 μm, Zoom 10 μm.
**F** Heatmap of unsupervised clustering of RNA sequencing data from M2

macrophages treated with IgA from control (ctrl, $n = 4$) or HS patients ($n = 4$). Data
represents four independent experiments. **G** Gene ontology enrichment analysis of
transcripts upregulated in macrophages treated with HS IgA versus control IgA.
**H**–**J** Detection of proinflammatory cytokines TNFα (**H**), IL-6 (**I**), and IL-1β (**J**) in
supernatants of macrophages treated with control or HS-IgA or HS-IgG for 24 h.
**K** qPCR analysis of *RORC* expression and **L** detection of IL-17A in supernatants of
naïve T cells cultured for 4 days with conditioned media from macrophages
exposed to control or HS-IgA or HS-IgG. Data are shown as median ± range of five to
six independent experiments in duplicates in (**H**, **I**, **J**, **L**) and four independent
experiments in duplicates for (**K**). For the statistical analysis, the Kruskal-Wallis test
was used in (**H**–**L**). Source data are provided as a Source data file.

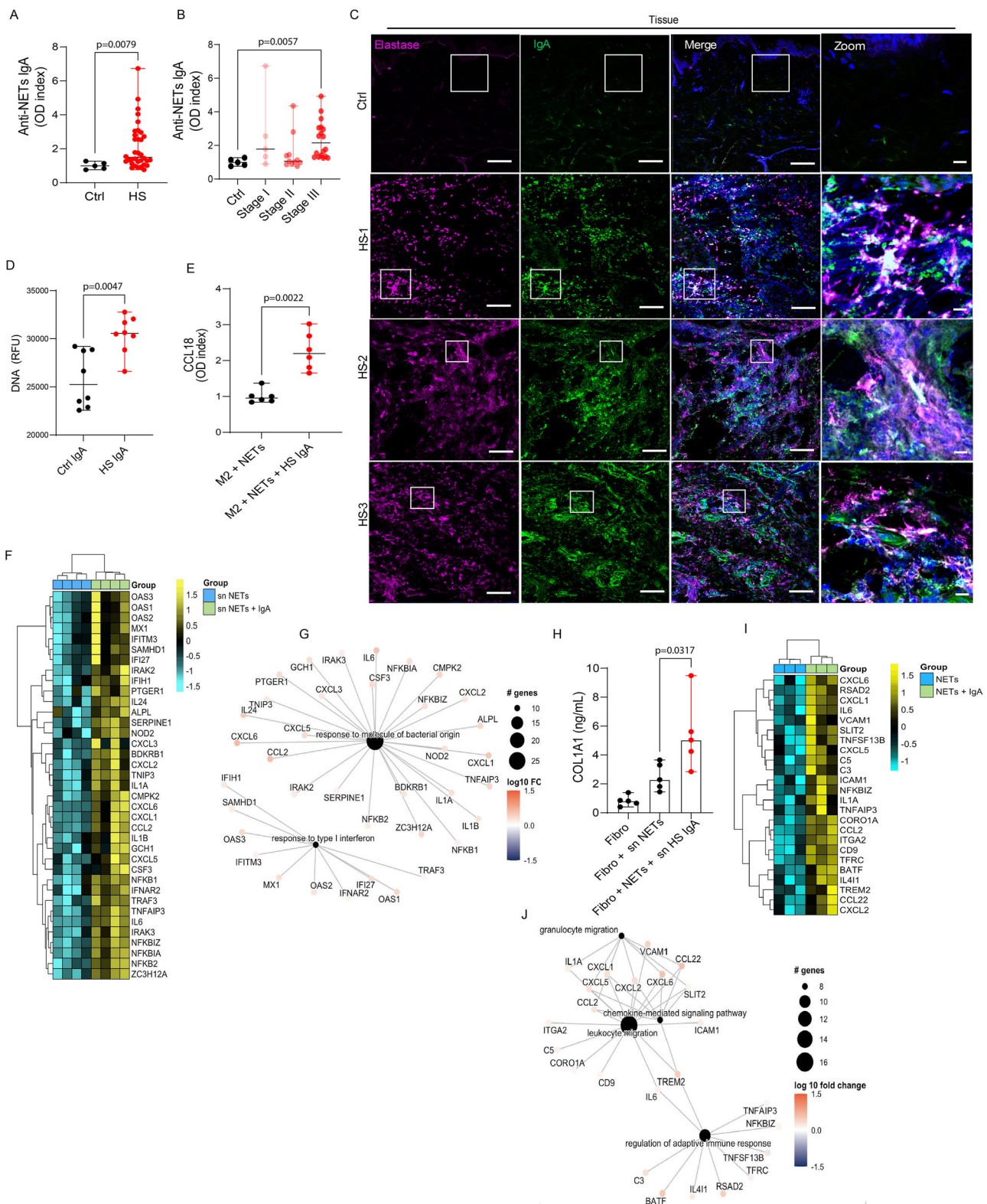

driving inflammation, while others might have regulatory or protective roles.

Notably, IgA autoantibodies against MDA5, complement proteins (C3/C4), ACE, and NLRP1 were enriched in advanced disease, implicating them in tissue damage and progression. Although IgA antibodies are often polyreactive, they are increasingly recognized as biologically relevant mediators of immune responses[22,26,27]. Further, the IgA detected in HS skin lesions did not exhibit reactivity against the

pathogens tested. While this finding argues against a dominant microbe-driven IgA response in our cohort, it does not exclude the possibility that IgA targeting other commensal or pathogenic microbes may be present in HS lesions, which warrants further investigation. Associations between IgA profiles and clinical features, including nodules, sinus tracts, acne, and comorbidities like diabetes, underscore the clinical significance of these responses. Interestingly, several IgA autoantibodies were inversely associated with anti-TNF therapy,

**Fig. 5 | NET-IgA immune complexes activate macrophages and skin fibroblasts.**
**A** Anti-NET IgA antibodies were measured in homogenized lysates from control
(Ctrl, $n = 5$) and HS patient samples ($n = 34$). **B** Levels of anti-NET IgA antibodies
stratified by HS severity (Ctrl, $n = 5$; Stage I, $n = 5$; Stage II, $n = 11$; Stage III, $n = 18$), as
defined by Hurley staging. Data represents three independent experiments in
duplicates. For the statistical analysis, a 2-sided unpaired Mann-Whitney U-test was
used in **A**, and a Kruskal-Wallis test was used in **B**. **C** Confocal images of three skin
lesions from HS patients and controls (Ctrl) stained for elastase (magenta), IgA
(green), and DNA (Hoechst, blue). Scale bars, 100 μm, Zoom 10 μm,
**D** Quantification of extracellular DNA following a 2-h incubation of neutrophils with
IgA from control ($n = 8$) or HS patients ($n = 8$). Results are the median ± range of
eight independent experiments in triplicate. RFU, relative fluorescence units.
**E** CCL18 production by macrophages after exposure to NETs alone or NETs

complexed with HS-derived IgA. Results are the median ± range of six independent
experiments in duplicates 2-sided unpaired Mann-Whitney U-test was used in (**D**, **E**).
**F** Heatmap showing unsupervised clustering of RNA sequencing data from skin
fibroblasts treated with supernatants from M2 macrophages exposed to NETs or
NETs + HS IgA ($n = 4$). **G** Gene ontology enrichment analysis of fibroblast tran-
scriptomes from panel (**F**). **H** Detection of COL1A1 in fibroblast supernatants after
24-h exposure to conditioned media from macrophages treated with NETs or
NETs + HS IgA. Results are the median ± range of five independent experiments in
duplicate. Statistical significance was assessed using the Kruskal-Wallis test. **I**
Heatmap showing unsupervised clustering of RNA sequencing data from skin
fibroblasts treated with NETs or NETs + HS IgA ($n = 3$). **J** Gene ontology enrichment
analysis of fibroblast transcriptomes from panel (**I**). Source data are provided as a
Source data file.

particularly adalimumab, raising the possibility that they may serve as
biomarkers for treatment response or resistance. In contrast, treat-
ment with infliximab, another anti-TNF agent, was positively asso-
ciated with the presence of IgA autoantibodies. This may reflect
infliximab's higher immunogenic potential due to its chimeric struc-
ture (approximately 75% human, 25% murine IgG1), which is known to
provoke stronger immune responses and autoantibody
production[21,28–31]. However, further studies are needed to fully eluci-
date the significance of these autoantibodies and to account for how
treatment itself might confound their interpretation.

Mechanistically, we demonstrate that IgA autoantibodies directly
contribute to immune activation. In particular, IgA2 binds skin-
resident cells such as keratinocytes, fibroblasts, and neutrophils,
forming immune complexes that activate mDCs via FcαRI. These
complexes promote antigen presentation and robust CD4+ T cell
responses with IFN-γ and TNF-α production, linking B cells, antigen-
presenting cells, and T cell–mediated inflammation. This mechanism
exemplifies how IgA bridges innate and adaptive immunity and rein-
forces the concept of an epithelial–immune axis in HS, similar to bona
fide systemic autoimmune diseases such as SLE and
dermatomyositis[32–36].

Functionally, IgA, but not IgG, also stimulates macrophages to
release IL-1β and IL-6, promoting Th17 differentiation. Interestingly,
IgG purified from the same HS skin lesion did not trigger Th17 polar-
ization, which is a key feature of HS and aligns with the immunologic
profile of other Th17-mediated diseases like psoriasis, Crohn's disease,
and ankylosing spondylitis[37–42]. Elevated IL-17A, IL-17F, and related
chemokines in HS support this shared pathogenic framework[37].

Macrophage polarization toward a profibrotic M2 phenotype
contributes to tissue remodeling and fibrosis in HS[43–45]. We show that
IgA antibodies in HS recognize and enhance NET formation, which in
turn bind more IgA, forming immune complexes. These NET–IgA
complexes activate macrophages to secrete CCL18, a key profibrotic
chemokine that drives fibroblast collagen production and contributes
to scarring and tunnel formation, as previously suggested[43,45]. Fibro-
blasts exposed to NET–IgA complexes adopt an inflammatory and
immune-competent phenotype, upregulating type I IFN–regulated
genes, and genes encoding for adhesion molecules, chemokines and
BAFF,further sustaining TLS stability and immune activation[46,47]. This
process may create a self-reinforcing loop wherein fibroblasts support
B cell survival and TLS architecture through chemokines like CXCL13,
CCL10, and CCL18, positioning them as key stromal regulators of
chronic inflammation in HS.

These findings expand the mechanistic framework of HS by
positioning IgA as an important mediator, linking adaptive auto-
immunity to innate immune activation and fibrosis. They also provide a
rationale for why conventional anti-TNF therapies may have variable
efficacy. Notably, several IgA autoantibodies showed inverse associa-
tions with anti-TNF therapy, raising the possibility that specific IgA
profiles could serve as biomarkers for treatment responsiveness or

therapeutic resistance. While B cell–depleting therapies like rituximab
have shown limited success, our data suggest that targeting TLS for-
mation, B-cell survival signals (e.g., BAFF, APRIL), or IgA effector
functions (e.g., FcαRI) may provide more durable disease control. The
inverse association between specific IgA autoantibodies and anti–TNF-
α therapy further supports the relevance of B cell–driven responses in
HS. Our findings align with emerging evidence that fibroblasts in HS
are not passive structural cells but active architects of TLS formation[15].
In response to immune cytokines, HS fibroblasts express CXCL13 and
CCL19, key chemokines that promote lymphocyte aggregation
through TNF-α–dependent feedback loops with B and T cells[15]. Nota-
bly, early TNF-α blockade can disrupt this fibroblast-driven TLSs, fur-
ther supporting the therapeutic potential of interfering with TLS
architecture.

Although this study provides robust evidence for local IgA
production and diverse IgA autoantibody activity in HS lesions,
several important limitations remain. First, the sample size, while
sufficient for exploratory and correlative analyses, may not fully
capture the heterogeneity of the HS patient population. These find-
ings should be validated in larger, ethnically diverse cohorts. Addi-
tionally, the analysis was limited to the antigens included in the
assays, thus the presence of IgA antibodies targeting other proteins
such as those from pathogens or other sources should be investi-
gated. While functional assays demonstrated that IgA activates key
immune pathways, other potential mechanisms remain to be eval-
uated. Further, while our data support a spatial association between
TLS and IgA-producing cells, we acknowledge that they do not defi-
nitively establish TLS as the sole or primary site of IgA production.
The cross-sectional design also limits interpretation of whether IgA
autoantibody responses are drivers or byproducts of chronic
inflammation and whether these responses are sustained over time.
Indeed, longitudinal studies tracking IgA profiles over time, including
during disease flares and treatment, are needed to clarify their role in
disease progression. Finally, in vivo validation using relevant disease
models will be critical to confirm the pathogenic potential of specific
IgA autoantibodies and to assess their value as therapeutic targets or
clinical biomarkers.

In summary, our study supports the redefinition of HS as a
chronic, immune complex–driven disease with an autoimmune com-
ponent centered on local immunoglobulin responses. Functional TLSs
within lesional skin drive B cell activation, IgA class switching, and
production of pathogenic autoantibodies. These antibodies form
immune complexes that activate neutrophils, macrophages, mDCs,
and fibroblasts, fueling a cycle of Th1/Th17 inflammation and fibrosis.
Targeting IgA pathways, TLS architecture, or B cell survival factors
offer promising therapeutic strategies to disrupt this pathogenic loop.
Future research should focus on the triggers of IgA class switching,
including the local microbiome, TLS regulation, and the therapeutic
potential of modulating IgA-mediated immunity in HS and related
inflammatory diseases.

## Materials and methods

### Collection of tissue samples

Healthy controls were recruited through advertisements, and written informed consent was obtained from all healthy volunteers. HS samples were obtained under protocols NA_00013177 and NA_00031269, approved by the Johns Hopkins University Institutional Review Board (IRB), IRB-19-MED-50 approved by the Howard University Office of Regulatory Research Compliance, and IRB-17-1506 approved by the University of North Carolina (UNC) Institutional Review Board. Lesional skin samples from HS patients were collected through the UNC Department of Dermatology's HS Program for Clinical and Research Excellence (ProCARE), following written informed consent. Skin samples were obtained during standard-of-care surgical procedures using a 6 mm punch biopsy. Clinical assessments were conducted by the surgeon or investigator at the time of surgery, documenting lesion type, regional and global lesion counts, current medications, and other relevant clinical and demographic information from electronic medical records or clinical registries. Additional lesional and non-HS samples were obtained from the Johns Hopkins tissue bank, sourced from patients undergoing HS lesion excision. All diagnoses were confirmed through clinical, visual, and histopathological evaluation. Collected tissues were stored at −80 °C until further analysis. All participants provided written informed consent.

### Isolation of protein from tissue and cells

Protein extraction from tissues or cells was performed as previously described[48]. Briefly, pulverized tissue samples were resuspended in RIPA buffer supplemented with a protease inhibitor cocktail (Roche). Following a 1-h incubation at 4 °C, the samples were centrifuged at $16,000 \times g$ for 10 min. The resulting supernatants were collected and transferred to fresh Eppendorf tubes. Protein concentrations were determined using the BCA Protein Assay Kit (Thermo Fisher), following the manufacturer's instructions.

### Antibodies

The following monoclonal antibodies were used: mouse anti-human IgA2-biotin (1:10,000, SouthernBiotech, cat no. 9140-08), mouse anti-human IgA1 (1:10,000, SouthernBiotech, cat no. 29130-08), mouse anti-human CD89 (1:100, Bio-Rad, cat no.MCA1824), mouse anti-human CD80 (1:100, Bio-Rad, cat no. MCA2071GA), mouse anti-human CD86 (1:100, Bio-Rad, cat no. MCA1118), and mouse anti-human GAPDH (clone 6C5; 1:1000, Thermo Fisher Scientific, cat no. AM4300). Antibodies were used for Western blotting according to manufacturer-recommended dilutions.

### NET isolation

NETs were isolated as previously described[48]. In brief, neutrophils from HS patients were resuspended in RPMI medium without phenol red and seeded into 24-well tissue culture plates. The cells were incubated at 37 °C for 3 h. Following incubation, supernatants were removed, and the adherent NETs were digested using micrococcal nuclease (10 U/mL; Thermo Fisher Scientific) in RPMI for 15 min at 37 °C. The resulting supernatants were collected and centrifuged at $2800 \times g$ for 5 min at 4 °C. NET-containing supernatants were then transferred to Eppendorf tubes and stored at −80 °C until further use.

### Anti-NET, macrophage and CD68 IgA detection

A 96-well plate was coated overnight at 4 °C with one of the following antigens diluted in PBS[14]: 10 μg/mL of spontaneously generated neutrophil extracellular traps (NETs), 2.5 μg/mL recombinant CD68 (antibodies-online, cat no. ABIN7092757), or 10 μg/mL macrophage lysate. Following coating, wells were blocked with 1% BSA for 1 h at room temperature. One microgram of total protein extracted from control or HS skin samples was added to each well and incubated overnight at 4 °C. After washing, HRP-conjugated goat anti-human IgA secondary antibody (1:5000; Life Technologies, cat no. a18781) was incubated for 1 h at room temperature. ELISA was developed using TMB substrate, and absorbance was measured at 450 nm using a Synergy HT microplate reader (Bio-Tek, Rockville, MD). Results are expressed as the optical density (OD) index, calculated as the ratio of the sample OD to the mean OD of healthy controls.

### NET quantification plate assay

To quantify NET formation, neutrophils were resuspended in RPMI without phenol red medium containing 0.2 μM Sytox Green (Invitrogen)[14]. Cells $(1 \times 10^4)$ were seeded into wells of a plate previously coated with either control or HS IgA. Cells were incubated for 2 h at 37 °C. Extracellular DNA was quantified by measuring fluorescence using a BioTek Synergy H1 Hybrid Reader (excitation: 485 nm; emission: 520 nm). Results are expressed as relative fluorescence units (RFU).

### Quantitative PCR analysis

RNA was isolated from tissue sections and cells as previously describes[5]. Briefly, RNA was isolated using the Direct-zol RNA MiniPrep Kit (Zymo Research) according to the manufacturer's instructions. Total RNA (300–500 ng) was reverse- transcribed using iScript RT single-strand complementary DNA (cDNA) (Bio-Rad). qPCR was performed using the TaqMan Gene Expression Master Mix (Thermo Fisher Scientific), human GAPDH primers (Hs99999905_m1), and sequence-specific primers for IGHA1(Hs00733892_m1), JCHAIN (H00376160_m1), and RORC (Hs01076112_m1) were used. Fold difference was calculated using the DCt equation.

### RNA sequencing and analysis

Between 50 and 500 ng of total RNA were used for cDNA synthesis, with RNA quantity normalized across all samples per experiment. RNA integrity was evaluated by analyzing an aliquot of each sample on an Agilent TapeStation using the High Sensitivity RNA ScreenTape. cDNA libraries were prepared using either the NEBNext Poly(A) mRNA Magnetic Isolation Kit or the NEBNext rRNA Depletion Kit (New England Biolabs), followed by library construction with the NEBNext Ultra II RNA Library Prep Kit for Illumina (New England Biolabs). Library fragment size and quality were assessed using the Agilent TapeStation with D1000 ScreenTape. Library concentrations were quantified using the PicoGreen dsDNA assay (Thermo Fisher Scientific). Libraries were diluted to 3 nM, pooled, and sequenced on the Illumina Novaseq X platform. Raw sequencing data were converted to FASTQ format using *bcl2fastq* (v2.20.0). Adapter trimming was performed using *Trim Galore* (v0.64.7). Pair-end 50 bp reads were mapped to the human genome (hg38) using STAR2.7.8a as implemented in Partek Flow (version 11.0.24.0604) with default settings. Gene expression levels were quantified as FPKM (Fragments Per Kilobase of exon per Million mapped reads) using the "quantify to annotation model" (Partek E/M) based on ensemble transcript release 109. Differential gene expression was conducted with ANOVA with Partek Flow.

### Western blot

Proteins isolated from tissue, macrophages, keratinocytes (HaCaT), and skin fibroblasts were quantified using the BCA Protein Assay Kit (Thermo Fisher) following the manufacturer's protocol. Equal amounts of total protein were separated on 4–12% Bis-Tris gradient gels (Invitrogen), then transferred onto nitrocellulose membranes. Membranes were blocked with 10% BSA for 30 min at room temperature, followed by overnight incubation with primary antibodies at 4 °C. After washing three times with PBS containing 0.1% Tween-20 (PBS-T), membranes were incubated with (1:10,000) secondary antibodies conjugated to IRDye 680 or 800CW. Blots were visualized using the LI-COR Odyssey CLx imaging system.

## IgA autoantibody detection assay

The assay was performed as previously described[13,49]. Briefly, skin lysates from 45 HS patients and 10 controls were pretreated with DNase I and diluted 1:50. Samples were then hybridized to nitrocellulose membrane-coated slides printed in duplicate with a panel of autoantigens related to various autoimmune conditions, including systemic lupus erythematosus (SLE), systemic sclerosis, Sjogren's syndrome, idiopathic inflammatory myositis, and rheumatoid arthritis (RA) (available at: https://utsw.corefacilities.org/service_center/show_external/5596?name=utsw-microarray-immune-phenotyping-core-facility). Following hybridization and washing steps, bound antibodies were detected using Cy3-labeled anti-human IgA (1:2000; Jackson ImmunoResearch Laboratories, cat no. 109-165-011). Arrays were scanned using a GenePix 4400 A Microarray Scanner, and fluorescence intensities were analyzed with GenePix 7.0 software (Molecular Devices). The background signal (from PBS) was subtracted from each sample's average signal intensity. A signal-to-noise ratio (SNR) was calculated, and antibody scores were determined using the formula $\log_2 (SNR \times NFI + 1)$, as previously described[50]

## Frozen sections and immunofluorescence staining

Frozen tissue sections (25 μm) were methanol fixed and permeabilized for 10 min at 4 °C, followed by blocking with 10% bovine serum albumin (BSA) for 1 h at room temperature. Where indicated, sections were incubated overnight at 4 °C with primary antibodies, including anti-CD68 (1:200, Abcam, cat no. ab201340), elastase (1:200, Abcam, cat no. ab68672), and FITC-conjugated IgA antibodies (1:200, Sigma, cat no. F5259), all diluted in 5% BSA. Following incubation, slides were washed three times with PBS. Sections were then incubated with secondary antibodies Alexa Fluor 555-conjugated anti-rabbit IgG or Alexa Fluor 488-conjugated donkey anti-mouse IgG (1:400; Invitrogen, cat no. A-21428, A-21202) for 1 h at room temperature. Nuclei were counterstained with Hoechst 33342 (1:1000, Invitrogen, cat no. H3570) for 10 min. After five PBS washes, sections were mounted with ProLong Gold Antifade Mounting (Invitrogen, cat no. P36930) and coverslip. Images were acquired using a Zeiss LSM780 confocal laser scanning microscope.

## IgA and IgG isolation from skin lysates

IgA and IgG antibodies were purified from 100 μg of Chronic stage (II-III) HS or control skin lysates using Protein M and G beads (InvivoGen), respectively. Samples were incubated with Protein M or G beads overnight at 4 °C. After incubation, the beads were washed three times with binding buffer (Cytiva, Sweden). IgA and IgG were then eluted using elution buffer and immediately neutralized with neutralization solution (Cytiva, Sweden). Buffer exchange was performed using Microcon-10 centrifugal filters (Millipore) according to the manufacturer's instructions.

## IgA or IgG stimulation and macrophage activation

Human CD14+ monocytes were isolated from control PBMCs using positive selection (Miltenyi Biotec). CD14+ monocytes were cultured with M-CSF (50 ng/mL; PeproTech) for 5–7 days. For stimulation, 96-well plates were coated overnight at 4 °C with purified IgA or IgG (isolated from control or HS skin lysates) diluted in PBS. Plates were then washed twice with phosphate-buffered saline before seeding with macrophages. Differentiated macrophages were detached using Dissociation solution (StemCells), and 50,000 cells were seeded per well in 200 uL. Cells were incubated for 24 or 72 h, after which supernatants were collected and analyzed for TNF-α, IL-6, and IL-1β using commercial ELISA kits (Invitrogen).

## Skin fibroblast treatment

Primary human skin fibroblasts were obtained from Coriell and cultured in Dulbecco's Modified Eagle Medium (DMEM) supplemented with 10% fetal bovine serum (FBS), 1% penicillin-streptomycin, and 1% L-glutamine at 37 °C in a humidified incubator with 5% $CO_2$. For stimulation experiments, fibroblasts were seeded into 12-well plates that had been pre-coated for 24 h with 50 μg of NETs alone or NETs + HS-IgA. After 24 h of incubation, total RNA was extracted for RNA sequencing analysis. In a parallel experiment, fibroblasts were also treated for 24 h with 10% supernatant from macrophages previously stimulated with either NETs alone or NETs + HS-IgA. RNA was isolated for transcriptomic profiling by RNA sequencing, and culture supernatants were collected and analyzed for collagen type I alpha 1 (COL1A1) levels using ELISA.

## mDC and CD4 T cell co-culture

Myeloid dendritic cells (mDCs) and CD4+ T cells were isolated from peripheral blood mononuclear cells (PBMCs) obtained from healthy control donors using the Myeloid Dendritic Cell Isolation Kit (Miltenyi Biotec, cat no. 130-094-487) and the CD4+ T Cell Isolation Kit (Miltenyi Biotec, cat no. 130-096-533), respectively, following the manufacturer's instructions. To generate immune complexes, 96-well plates were pre-coated overnight at 4 °C with IgA purified from the HS skin lesion. Keratinocyte lysates were prepared and denatured by heating at 95 °C for 5 min, then incubated with the IgA-coated wells for 2 h at room temperature. Plates were washed twice with PBS to remove unbound proteins. Isolated mDCs and CD4+ T in a ratio of 1:2 cells were then co-cultured in the presence or absence of keratinocyte-IgA immune complexes, with or without the addition of blocking antibodies against CD89 (1:100, Bio-Rad, cat no. MCA1824), CD80 (1:100, Bio-Rad, cat no. MCA2071GA), or CD86 (1:100, Bio-Rad, cat no. MCA1118) for 4–5 days. Culture supernatants were collected and analyzed for IFN-γ and/or TNF-α production.

## Th17 polarization

Naïve CD4+ T cells were isolated from peripheral blood mononuclear cells (PBMCs) using the Naïve CD4+ T Cell Isolation Kit II, human (Miltenyi Biotec, cat no. 130-094-131), according to the manufacturer's protocol. Isolated naïve T cells were activated with CD3/CD28 ImmunoCult antibodies (1:40, Stem Cell, cat no.10971) and cultured in the presence of 10% of supernatant from macrophages that had been previously exposed to either control or HS purified IgA or IgG. Cells were incubated at 37 °C in a humidified incubator with 5% $CO_2$ for 4 days. Following incubation, total RNA was extracted for qPCR analysis of the Th17 lineage-defining transcription factor *RORC*. Culture supernatants were analyzed for IL-17A levels using ELISA to assess Th17 polarization.

## Statistical analysis

All statistical analyses were performed using GraphPad Prism version 9.5.0 (GraphPad Software, La Jolla, CA). For comparisons involving non-normally distributed data, the Mann–Whitney U test was applied. Group comparisons were conducted using the Kruskal–Wallis test followed by Dunn's multiple comparison test. Pearson correlation was used to assess associations between continuous variables. A *p*-value of <0.05 was considered statistically significant.

## Data availability

RNA sequencing data from this study are available under the GEO accession number GSE298444. [https://www.ncbi.nlm.nih.gov/geo/query/acc.cgi?acc=GSE298444]. Source data are provided with this paper.

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

## Acknowledgments

This study was supported [in part] by the Intramural Research Program of the National Institutes of Health (NIH), NIAMS/NIH, ZIA AR041199 (MJK), the Skin of Color Society-Career Development Award (CC-R) and NIH grants 1R21AR075996 and 1R01AR083790-01A1 (CJS). The contributions of the NIH author(s) were made as part of their official duties as NIH federal employees, are in compliance with agency policy requirements, and are considered Works of the United States Government. However, the findings and conclusions presented in this paper are those of the author(s) and do not necessarily reflect the views of the NIH or the U.S. Department of Health and Human Services.

## Author contributions

C.C.-R., E.P.-M., and W.A. performed the experiments; C.C.-R., L.J.O., E.P.-M., N.H., and K.J. analyzed the data and performed statistical analyses; W.A., T.M., A.K.Z., G.A.O., A.S.B., and C.J.S. provided specimens, drafted the manuscript, and clinical insight and information; M.J.K. reviewed and revised the manuscript; C.C.-R. was involved in overall design, conceptualization and/or manuscript preparation.

## Competing interests

ASB is a consultant for Senté, Inc. GAO is on an Advisory Board for Pfizer, UCB, Eli Lilly, and Novartis and a consultant for Janssen. CJS serves as Secretary of the Board of Directors for HS Foundation, speaker for AbbVie, Novartis and UCB, a consultant for AbbVie, Novartis, UCB, Sanofi, Incyte, InfaRx, Astrazeneca, Navigator Medicines, Moonlake Therapeutics and Elasmogen, and an investigator for Novartis, UCB, Incyte, InflaRx and Astrazeneca. All other authors declare no conflict of interest.

## Additional information

**Carmelo Carmona-Rivera** ⑩[1] ✉, **Liam J. O'Neil**[2], **Eduardo Patino-Martinez**[1], **William G. Ambler**[1], **Teja Mallela**[3], **Norio Hanata** ⑩[1], **Arsema K. Zadu**[4], **Kan Jiang** ⑩[5], **Ginette A. Okoye**[4,6], **Angel S. Byrd**[4,6], **Christopher J. Sayed**[3] & **Mariana J. Kaplan** ⑩[1]

[1]Systemic Autoimmunity Branch, National Institute of Arthritis and Musculoskeletal and Skin Diseases, National Institutes of Health, Bethesda, MD, USA. [2]Manitoba Centre for Proteomics and Systems Biology, Department of Internal Medicine, University of Manitoba, Winnipeg, MB, Canada. [3]Department of Dermatology, University of North Carolina, Chapel Hill, NC, USA. [4]Department of Dermatology, Howard University College of Medicine, Washington, DC, USA. [5]Biodata Mining and Discovery Section, National Institute of Arthritis and Musculoskeletal and Skin Diseases, National Institutes of Health, Bethesda, MD, USA. [6]Department of Dermatology, Johns Hopkins University School of Medicine, Baltimore, MD, USA. ✉e-mail: carmelo.carmona-rivera@nih.gov

