## [Transparent Peer Review file · Nature Communications]

IgA Autoantibodies Promote inflammation, Th17 Polarization, and Fibrotic Responses in Hidradenitis Suppurativa

Corresponding Author: Dr Carmelo Carmona-Rivera

Version 0:

Reviewer comments:

Reviewer #1

(Remarks to the Author)

Hidradenitis suppurativa (HS) is a relatively common inflammatory skin disease with potentially devastating impacts on patient quality of life. Mechanisms of disease pathogenesis remain poorly defined. In their manuscript Carmona-Rivera et al., build on recent findings that B cells and supporting tertiary lymphoid structures are highly active in HS to comprehensively detail the functional role of lesional auto-IgA. The current work identifies high enrichment of auto-IgA in lesional HS skin samples and maps which specific antigens are targeted including correlations with clinical parameters. A number of strong ex vivo experiments with IgA isolated from lesional skin are performed to demonstrate that these antibodies are functionally relevant for regulating antigen presentation to T cells, macrophage activation, and fibroblast activity. Altogether the findings present a strong new line of evidence that local autoimmunity contribute to the inflammatory pathogenesis of HS. The manuscript is well written and robust experiments/analyses are presented. Findings have important implications for HS but also may inform our thinking about the link between autoimmunity and inflammatory disease more generally. Therefore, I believe the manuscript is well suited for the broad interest readership at Nature Communications. I have minor comments listed below:

-In Figure 1, total IgA gene expression and protein levels present in lesional skin are quantified. As done with most other figures it would be helpful to correlate this by Hurley stage given this in of itself could be a biomarker.

-At the end of the first results section "IgAs are present in HS skin lesions" it is stated that the "findings indicate TLS support local differentiation of IgA-producing cells..." while this is logical, the presented data in of themselves do not test this hypothesis. I would be more comfortable if the sentence was deleted or amended to remove reference to the role of TLS.

-The image quality of Fig 2A,B,D is low and I had trouble reading the individual antigens/parameters.

-The correlation analysis in Fig 2D is important but as a comparator it would be helpful to also include total IgA.

-In Figure 4c the CD4/mDCs are from healthy donors but the figure legend wording could be misconstrued as indicating the cells are isolated from HS patients.

-Fig 5f is slightly cut off.

Reviewer #2

(Remarks to the Author)

Overall Assessment

This manuscript investigates the role of IgA autoantibodies in hidradenitis suppurativa, demonstrating their ability to activate macrophages, promote Th17 polarization, and contribute to fibrotic responses. The topic is timely and important, linking humoral immunity to key inflammatory and tissue-remodeling processes in HS. The study is original and clinically relevant, but several methodological and interpretative weaknesses limit the strength of the conclusions.

Strengths

1. Originality: The potential role of IgA autoantibodies in HS has been scarcely studied, making this manuscript highly innovative.
2. Integrated approach: The authors connect autoantibodies, macrophage activation, Th17 polarization, and fibrosis within a unified mechanistic framework.

Weaknesses

IgA specificity

- IgA autoantibodies are often polyreactive and prone to nonspecific binding. The broad reactivity observed in Figures 2A and 2B might reflect nonspecific interactions rather than true antigen-specific responses. This issue should be discussed.
- If IgA are indeed polyreactive, they may target microbial as well as self-antigens. This possibility should be considered, especially given the suspected role of dysbiosis in HS.
- Since many HS patients receive antibiotics, it would be valuable to explore whether IgA responses are modulated in this context. This could help disentangle whether the observed activity is linked to microbial exposure or autoreactive.

Variability in IgA immunostaining

The in situ IgA staining shown in Figures 1G, 4C, and 5C appears inconsistent. Quantification or more standardized representative images would strengthen confidence in these findings.

Origin of IgA production within TLS

It remains unclear whether TLS are the main sites of IgA production. Demonstrating or discussing this point would significantly enhance the mechanistic interpretation.

Comparison with IgG responses is missing

In chronic inflammation, isotype switching typically favors IgG rather than IgA. It would be highly informative to quantify IgG responses in parallel, to assess whether IgG autoantibodies might contribute even more significantly to macrophage activation, Th17 polarization, or NET recognition. Without this comparison, the specific contribution of IgA remains difficult to establish.

Requirement for immune complexes

In figure 3, it is unclear whether IgA alone can activate myeloid dendritic cells (mDCs), or whether immune complex formation is required. Clarification is essential, since Fc α R signaling usually depends on multivalent engagement.

Impact of IgA on M2 macrophages

It would be highly relevant to test whether HS-derived IgA alters the phenotype of M2 macrophages. For example, do they produce fibrotic mediators such as CCL18, or undergo transcriptomic changes consistent with a pro-fibrotic program?

Version 1:

Reviewer comments:

Reviewer #1

(Remarks to the Author)

The authors have thoughtfully and carefully addressed my initial comments/concerns. My strong enthusiasm for this manuscript remains as I believe it applies a rigorous approach to dissecting the potential pathogenic role of lesional antibody (especially IgA) in HS. Conclusions are well supported by the data presented and these data are highly important for the field.

A few small errors should be addressed:

- In figure S1A the split axis cuts off some of the data points
- Line 62: it's misleading to shorten follicular dendritic cells to "DCs" as they are mesenchymal in origin. Use FDC
- Line 315: Is this a typo?
- Line 326: typo "BAFF ,"

Reviewer #2

(Remarks to the Author)

The revised manuscript has been substantially improved following the initial review. The authors have addressed all of my comments carefully and comprehensively, providing additional data, analyses, and clarifications that significantly strengthen both the rigor and the mechanistic interpretation of the study.

In particular, the added analyses examining IgA reactivity against microbial antigens, as well as the direct functional comparison between IgA and IgG derived from the same lesional samples, meaningfully reinforce the central conclusions. The clarification regarding immune complex requirements and the tempered interpretation of TLS involvement further enhance the scientific precision of the manuscript.

Overall, the revised version presents a coherent and well-supported framework linking lesional IgA autoantibodies to

macrophage activation, Th17 polarization, and fibrotic responses in hidradenitis suppurativa. The manuscript is now considerably strengthened.

RESPONSE TO REVIEWERS' COMMENTS

Reviewer #1 (Remarks to the Author):

Hidradenitis suppurativa (HS) is a relatively common inflammatory skin disease with potentially devastating impacts on patient quality of life. Mechanisms of disease pathogenesis remain poorly defined. In their manuscript Carmona-Rivera et al., build on recent findings that B cells and supporting tertiary lymphoid structures are highly active in HS to comprehensively detail the functional role of lesional auto-IgA. The current work identifies high enrichment of auto-IgA in lesional HS skin samples and maps which specific antigens are targeted including correlations with clinical parameters. A number of strong ex vivo experiments with IgA isolated from lesional skin are performed to demonstrate that these antibodies are functionally relevant for regulating antigen presentation to T cells, macrophage activation, and fibroblast activity. Altogether the findings present a strong new line of evidence that local autoimmunity contribute to the inflammatory pathogenesis of HS. The manuscript is well written and robust experiments/analyses are presented. Findings have important implications for HS but also may inform our thinking about the link between autoimmunity and inflammatory disease more generally. Therefore, I believe the manuscript is well suited for the broad interest readership at Nature Communications. I have minor comments listed below:

We thank the reviewer for this thoughtful and comprehensive evaluation of our work. We appreciate the positive assessment of the novelty, experimental rigor, and broader implications of our findings in HS and other chronic inflammatory diseases.

-In Figure 1, total IgA gene expression and protein levels present in lesional skin are quantified. As done with most other figures it would be helpful to correlate this by Hurley stage given this in of itself could be a biomarker.

We thank the reviewer for this helpful suggestion. We have now incorporated the distribution of total IgA gene expression and protein levels stratified by Hurley stage. These results are now presented in Supplementary Figure S1.

-At the end of the first results section “IgAs are present in HS skin lesions” it is stated that the “findings indicate TLS support local differentiation of IgA-producing cells...” while this is logical, the presented data in of themselves do not test this hypothesis. I would be more

comfortable if the sentence was deleted or amended to remove reference to the role of TLS.

We thank the reviewer for this important point and agree that the original wording could be interpreted as overextending beyond the data initially presented. To address this concern, we have revised the text to more accurately reflect the experimental evidence.

In the revised manuscript, we have added immunostaining for IgA producing cells in conjunction with B-cell markers in HS skin, demonstrating the presence of IgA-expressing B cells within cellular aggregates resembling tertiary lymphoid structures (Figure 1J). These data, together with our qPCR findings showing increased IgA transcripts in lesional compared with non-lesional skin, and the detection of IgA in HS skin lesions but not in the matched serum samples, support the presence of locally differentiated IgA-producing cells in the skin.

We have revised the Results text to avoid implying a definitive mechanistic role of TLS structures in IgA class switching and instead describe the findings as evidence consistent with local IgA production. We believe these changes appropriately align our conclusions with the scope of the experimental data.

-The image quality of Fig 2A,B,D is low and I had trouble reading the individual antigens/parameters.

We apologize for the lack of clarity and agree that the heatmaps in Fig. 2A, B, and D contain a large number of antigens, which made them difficult to read at the original resolution. To address this, we have increased the figure size and optimized the image resolution. When viewed at higher magnification, individual antigens and expression patterns are now clearly discernible.

-The correlation analysis in Fig 2D is important but as a comparator it would be helpful to also include total IgA.

We thank the reviewer for this request. In response, we have performed an additional correlation analysis using total IgA and IgG levels measured in HS skin lesions. These results are now included in Figure 2F.

-In Figure 4c the CD4/mDCs are from healthy donors but the figure legend wording could be misconstrued as indicating the cells are isolated from HS patients.

To clarify, the mDC-CD4⁺ T cell experiments shown in Figure 3 were performed using mDCs and CD4⁺ T cells isolated from healthy donors and cultured in the presence of IgA purified

from HS patients. The corresponding figure legend reads: “IgA facilitates keratinocyte antigen uptake by mDCs via FcαR and promotes CD4⁺ T cell activation.”

-Fig 5f is slightly cut off.

Thanks for noticing this; we have now corrected this.

Reviewer #2 (Remarks to the Author):

Overall Assessment

This manuscript investigates the role of IgA autoantibodies in hidradenitis suppurativa, demonstrating their ability to activate macrophages, promote Th17 polarization, and contribute to fibrotic responses. The topic is timely and important, linking humoral immunity to key inflammatory and tissue-remodeling processes in HS. The study is original and clinically relevant, but several methodological and interpretative weaknesses limit the strength of the conclusions.

We thank the reviewer for recognizing the novelty, clinical relevance, and potential impact of our study linking IgA autoantibodies to key inflammatory and tissue-remodeling processes in HS. We also appreciate the constructive feedback regarding methodological and interpretative aspects and have carefully addressed each of the points raised to strengthen the rigor and clarity of our conclusions.

Strengths

1. Originality: The potential role of IgA autoantibodies in HS has been scarcely studied, making this manuscript highly innovative.

We thank the reviewer for highlighting the originality and integrative nature of our study.

2. Integrated approach: The authors connect autoantibodies, macrophage activation, Th17 polarization, and fibrosis within a unified mechanistic framework.

We are pleased that the innovative aspect of investigating IgA autoantibodies in HS and our approach linking humoral immunity to macrophage activation, Th17 polarization, and fibrotic responses are recognized. We consider that these insights provide a valuable

framework for understanding the multifaceted mechanisms driving HS pathogenesis.

Weaknesses

IgA specificity

- IgA autoantibodies are often polyreactive and prone to nonspecific binding. The broad reactivity observed in Figures 2A and 2B might reflect nonspecific interactions rather than true antigen-specific responses. This issue should be discussed.

We appreciate the reviewer's comment regarding IgA polyreactivity. While IgA antibodies, particularly those arising in mucosal and tissue compartments, are known to exhibit a degree of polyreactivity, this property does not necessarily imply nonspecific or artifactual binding. Indeed, accumulating evidence indicates that IgA polyreactivity can be biologically meaningful and functionally relevant (Bunker et al., *Science* 2018; Lindner, *J Exp Med* 2012; Barone et al., *Gastro* 2011; Kabbert et al., *J Exp Med* 2020). In our study, the reproducible and selective downstream responses following IgA engagement—including induction of IL-1 β production by macrophages and preferential Th17 polarization—argue against nonspecific interactions as the primary explanation for the broad reactivity patterns observed in Figures 2A-B. We therefore interpret these data as reflecting biologically relevant IgA-antigen interactions, while acknowledging that IgA polyreactivity may contribute to the breadth of antigen recognition. We have added text to the Discussion to clarify this point and to appropriately contextualize the findings.

- If IgA are indeed polyreactive, they may target microbial as well as self-antigens. This possibility should be considered, especially given the suspected role of dysbiosis in HS.

We appreciate the reviewer's comment regarding IgA polyreactivity and the possibility that IgA antibodies in HS may target both microbial and self-antigens, particularly in the context of dysbiosis. To address this, we assessed IgA reactivity against several microbes implicated in HS or commonly present on the skin, including *E. coli*, *Lactobacillus casei*, and *P. gingivalis*. We did not detect significant differences in IgA reactivity to these microbial antigens between healthy skin and HS lesions (Figure 2E, F; Supplementary Figure S2).

These findings suggest that, for the microbial antigens examined, IgA responses do not differ between disease and control tissue and are unlikely to account for the increased self-reactive IgA observed in HS lesions. Moreover, IgA reactivity to these microbial targets did not confound the associations observed between IgA self-reactivity and clinical parameters. While IgA polyreactivity remains a theoretical consideration, our data indicate that IgA

targeting of these microbial antigens is unlikely to be the primary driver the self-reactive IgA signatures identified in HS skin.

- Since many HS patients receive antibiotics, it would be valuable to explore whether IgA responses are modulated in this context. This could help disentangle whether the observed activity is linked to microbial exposure or autoreactive.

We agree that antibiotic use is an important variable to consider when interpreting IgA responses in HS. To address this, we performed correlation analyses stratifying patients based on treatment with topical clindamycin or doxycycline. We did not observe significant correlations between antibiotic exposure IgA levels in HS skin (clindamycin $r = 0.2271$, $p = 0.1678$; doxycycline $r = 0.3780$, $p = 0.1004$) nor with IgG levels (clindamycin $r = 0.1325$, $p = 0.2889$; doxycycline $r = 0.3382$, $p = 0.1147$). If local IgA or IgG accumulation were primarily driven by microbial exposure, reduced antibody levels would be expected in patients receiving antibiotics. The absence of such associations suggests that the observed antibody responses are more likely autoreactive rather than directly linked to microbial burden. We acknowledge that the relatively small cohort size and imbalance between treated and untreated patients limit definitive conclusions. Accordingly, indirect or context-dependent microbial contributions cannot be entirely excluded and warrant further investigation in larger, more balanced cohorts.

Variability in IgA immunostaining

The in situ IgA staining shown in Figures 1G, 4C, and 5C appears inconsistent.

Quantification or more standardized representative images would strengthen confidence in these findings.

Thank you for the comment. In response, we repeated immunofluorescence for IgA, NETs, and macrophages in both control and HS skin samples. We have replaced Figures 1G, 4C, and 5C with more standardized and representative images that better illustrate the distribution and co-localization of these components across samples. These updated images more accurately reflect the reproducibility of our observations.

Origin of IgA production within TLS

It remains unclear whether TLS are the main sites of IgA production. Demonstrating or discussing this point would significantly enhance the mechanistic interpretation.

To address this point, we have added confocal images of HS skin lesions demonstrating B cell aggregates consistent with tertiary lymphoid structures (TLS) and the presence of IgA-positive cells within or adjacent to these structures in Figure 1J. While these data support a spatial association between TLS and IgA-producing cells, we acknowledge that they do not definitively establish TLS as the sole or primary sites of IgA production. We have therefore tempered our interpretation accordingly and discussed this limitation in the revised manuscript.

Comparison with IgG responses is missing

In chronic inflammation, isotype switching typically favors IgG rather than IgA. It would be highly informative to quantify IgG responses in parallel, to assess whether IgG autoantibodies might contribute even more significantly to macrophage activation, Th17 polarization, or NET recognition. Without this comparison, the specific contribution of IgA remains difficult to establish.

We thank the reviewer for this insightful comment. In response, we have now quantified total IgG levels in the same HS skin samples analyzed for IgA (Supplementary Figure S1). Notably, IgG and IgA levels were comparable, indicating that IgA accumulation in HS lesions is not simply secondary to a dominant IgG response.

To further delineate functional differences, we purified IgA and IgG from the same HS tissue samples and compared their effects on macrophage activation. Both isotypes induced IL-6 and TNF production at similar or greater levels following IgG stimulation. However, IL-1 β production differed significantly between IgA- and IgG-stimulated macrophages (Figures 4H–J). Supernatants from these macrophage cultures were then used in Th17 polarization assays. While supernatants from IgG-stimulated macrophages failed to promote Th17 polarization, those from IgA-stimulated macrophages robustly induced Th17 polarization (Figure 4L). Together, these findings suggest that although IgG contributes to inflammatory cytokine production, IgA uniquely promotes a macrophage-TH17 axis, supporting a nonredundant role for IgA in shaping pathogenic immune responses in HS.

Requirement for immune complexes

In figure 3, it is unclear whether IgA alone can activate myeloid dendritic cells (mDCs), or whether immune complex formation is required. Clarification is essential, since Fc α R signaling usually depends on multivalent engagement.

We have now included experiments in which mDCs were incubated with purified IgA alone as a control condition. These data are included in the revised Figure 3

Impact of IgA on M2 macrophages

It would be highly relevant to test whether HS-derived IgA alters the phenotype of M2 macrophages. For example, do they produce fibrotic mediators such as CCL18, or undergo transcriptomic changes consistent with a pro-fibrotic program?

We thank the reviewer for this insightful suggestion. To address this, we incubated M2 macrophages with purified IgA derived from either healthy control skin or HS lesional skin. Treatment with control IgA reduced CCL18 production, whereas HS-derived IgA did not significantly alter CCL18 levels compared with untreated M2 macrophages (Supplementary Figure S3). These findings indicate that HS IgA alone is insufficient to directly induce a pro-fibrotic macrophage phenotype.

Importantly, these results suggest that additional contextual signals, such as NET priming or IgA-NET immune complex formation, as demonstrated in Figure 5, may be required to drive fibrotic or tissue-remodeling programs. While our data does not support a direct pro-fibrotic effect of HS IgA in isolation, they highlight a multi-step mechanism in which IgA contributes to inflammation and tissue remodeling in concert with other immune factors. We believe this adds important mechanistic nuance to the role of IgA in HS pathogenesis.

RESPONSE TO REVIEWERS' COMMENTS

Reviewer #1 (Remarks to the Author):

The authors have thoughtfully and carefully addressed my initial comments/concerns. My strong enthusiasm for this manuscript remains as I believe it applies a rigorous approach to dissecting the potential pathogenic role of lesional antibody (especially IgA) in HS. Conclusions are well supported by the data presented and these data are highly important for the field.

We sincerely thank the reviewer for the thoughtful and encouraging feedback. We greatly appreciate your recognition of the rigor of our approach and the significance of our findings.

A few small errors should be addressed:

-In figure S1A the split axis cuts off some of the data points

We thank the reviewer for highlighting this issue. We have revised Figure S1A and ensured that all data points are fully visible.

-Line 62: it's misleading to shorten follicular dendritic cells to "DCs" as they are mesenchymal in origin. Use FDC

Thanks for the clarification, we have updated the revised version accordingly.

-Line 315: Is this a typo?

We have corrected accordingly.

-Line 326: typo "BAFF,"

We have corrected accordingly.

Reviewer #2 (Remarks to the Author):

The revised manuscript has been substantially improved following the initial review. The authors have addressed all of my comments carefully and comprehensively, providing additional data, analyses, and clarifications that significantly strengthen both the rigor and the mechanistic interpretation of the study.

In particular, the added analyses examining IgA reactivity against microbial antigens, as well as the direct functional comparison between IgA and IgG derived from the same lesional samples, meaningfully reinforce the central conclusions. The clarification regarding immune complex requirements and the tempered interpretation of TLS involvement further enhance the scientific precision of the manuscript.

Overall, the revised version presents a coherent and well-supported framework linking lesional IgA autoantibodies to macrophage activation, Th17 polarization, and fibrotic responses in hidradenitis suppurativa. The manuscript is now considerably strengthened.

We sincerely thank the reviewer for this thoughtful and highly encouraging assessment. We are pleased that these revisions have strengthened the rigor, mechanistic clarity, and overall framework of the manuscript.